# Effect of Content and Length of Polypropylene Fibers on Strength and Microstructure of Cementitious Tailings-Waste Rock Fill

**Bo Gao [1,2], Shuai Cao [1,2],\* and Erol Yilmaz [3,\***]

1   School of Civil and Resources Engineering, University of Science and Technology Beijing, Beijing 100083, China
2   State Key Laboratory of High-Efficient Mining and Safety of Metal Mines of Ministry of Education, University of Science and Technology Beijing, Beijing 100083, China
3   Department of Civil Engineering, Geotechnical Division, Recep Tayyip Erdogan University, Fener, Rize TR53100, Türkiye
\*   Correspondence: sandy_cao@ustb.edu.cn (S.C.); erol.yilmaz@erdogan.edu.tr (E.Y.)

**Abstract:** The mechanical strength properties of cemented tailings backfill are very important for the safe and environmentally friendly mining of mineral resources. To check the impact of polypropylene fiber on strength and microstructure of cementitious tailings waste rock fill (CTWRF), diverse fiber lengths (6 and 12 mm) and dosages (0-control specimen, 0.3, 0.6, and 0.9 wt.%) were considered to prepare fiber-reinforced CTWRF (FRCTWRF) matrices. Experiments such as UCS (uniaxial compressive strength), X-ray CT (computed tomography), and SEM (scanning electron microscopy) were implemented to better characterize the backfills studied. Results showed that UCS performance of FRCTWRF was the highest (0.93 MPa) value at 6 mm fiber long and 0.6 wt.% fiber content. The peak strain of FRCTWRF was the highest (2.88%) at 12 mm fiber long and 0.3 wt.% fiber content. Growing the length of fiber within FRCTWRF can reduce its fracture volume, enhancing the crack resistance of FRCTWRF. Fiber and FRCTWRF are closely linked to each other by the products of cement hydration. The findings of this work will offer the efficient use of FRCTWRF in mining practice, presenting diverse perspectives for mine operators and owners, since this newly formed cementitious fill quickens the strengths required for stope backfilling.

**Keywords:** fiber-reinforced cementitious tailings filling; fiber; strength; X-ray CT; microstructure

## 1. Introduction

To capably realize the green mining of underground ore resources [1], backfilling mining is one of the key mining techniques [2,3]. In recent years, scholars and engineers have commenced abundant researches on different subjects such as mining process [4], backfilling materials mechanics [5], and blasting parameter optimization [6]. During the mining and mineral processing sections, the two main types of solid wastes generated were waste rock [7] and minerals tailings [8], respectively. The storage of these wastes not only conquers large volumes of surface lands [9], but also leads to serious ecological hazards [10]. As the underground run of mine (ROM) ore was extracted, a large number of mined-out areas (i.e., stopes) were formed [11,12]. These stopes are essential for the overall stability of underground mining and must be managed well [13]. Hence, the ratio of the gap area created by the stopes to the total opening area in the underground mine should not be higher than 15% [14]. If these stopes are not backfilled in a timely manner [15], then stability problems, including local or regional collapses, may occur in mines [16], and even fatal mining accidents may occur [17]. The way to prevent all these bad consequences is to ensure the stability of each gap opened by backfilling without wasting time [18]. The most common type of the backfills used in mines is cementitious tailings/paste backfill

(CTB/CPB: [19–21]) since it allows for fast settling [22], easy operation [23], offering local/regional support [24], and placing a key part of tailings underground [25], reducing the aboveground environmental problems [26]. Indeed, this is vital for sustainable/efficient mining [27]. The comprehensive utilization of both waste rock and processing tailings for back-filling the mined-out openings or stopes [28] could not only prevent the extraction area collapse [29] but also reduce the stockpiling of solid waste on the surface nearby the mine site [30,31].

CTB/CPB is a category of composites that is basically manufactured by a blend of mining waste, cement, and water [32–35]. The sums of these materials used in the matrix can vary significantly depending on the function of the backfill created for underground mines [36,37]. To attain the required mechanical strengths of CTB while maintaining its rheological properties [38], the proportion of each component must be determined [39]. Various extra materials such as sand [40,41], fiber [42,43], and pozzolans [44,45] have been recently used to reduce the backfill costs while increasing its strength behavior. Since ROM ore is also crushed into finer particles to boost ore recovery [46], the tailings employed for CTB making can easily cause low UCS values [47]. CTB with coarse-sized tailings has larger porosity than CTB with fine-sized tailings, resulting a major reduction in UCS [48]. Few studies are focused on CTB having other reinforcing materials such as fiber to improve their mechanical strengths [49–51]. Accordingly, improving the strength performance of CTB remains a chief research topic.

For filling materials, researchers have started important lab/field investigations on uniaxial-triaxial compression [52,53], Brazilian splitting [54], three-/four-point bending [55], and shearing [56] tests. The research materials involved CTB [57,58], cementitious coal fly ash/coal gangue backfill materials [59,60], and others [61]. In terms of research tools, many scholars also considered using acoustic emission (AE) system [62], industrial computed tomography (CT) scan [63], mercury intrusion porosimetry [64], and nuclear magnetic resonance system [65]. Yang et al. [66] examined the crack behavior of fiber-reinforced CTB by means of an AE system. Results showed that b-value and AE fractal aspect could assess the evolution of cracks. Mashifana and Sithole [67] assumed that the size distribution of the particles tested could govern the mechanical strength behavior of CTB specimens. Findings revealed that the slighter the grain size, the higher the strength of filling specimens. Xue and Yilmaz [68] found that polypropylene fibers were effective in improving CTB's strength behavior. Sun et al. [69] observed that adding waste rock can develop the backfill's strength. Results indicated that CTBs with 25–30% waste rock had better flowability than others. Some researchers [70,71] also quantified that fibers were able to importantly inhibit the rapid extension of the cracks observed within CTB specimens based on SEM micrograph observations.

This work firstly aims to investigate the impact of length and content of polypropylene fiber (PF) on quality and properties of fiber-reinforced cemented tailings-waste rock fill (FRCTWRF). Secondly, it aims to define the mechanical (uniaxial strength/peak strain) and microstructure (crack volume) behavior of FRCTWRF through UCS, X-ray CT, and SEM observations. Regarding the novelty of this work, it is significantly original, since it involved analyzing PF-reinforced CTWRF samples under elevated temperatures through SEM and micro-CT scan systems.

## 2. Materials and Methods

### 2.1. Materials

5–7 mm sized waste rock and gold tailings were utilized for making CTB samples. ordinary Portland cement (OPC) 42.5R was employed as a basic cementing source. Tap water in the lab was used as mixing water. A laser diffraction particle size analyzer was utilized to characterize specimens' grain size distribution (GSD). GSD profiles of gold both tailings and cement are shown in Figure 1. Note that GSD plot of gold tailings was similar to that of cement. 10% of the tailings were less than 2 μm, and 50% of the tailings were less

than 20 μm. The curvature coefficient of the tailings was 2.07, and the uniformity coefficient was 20.98.

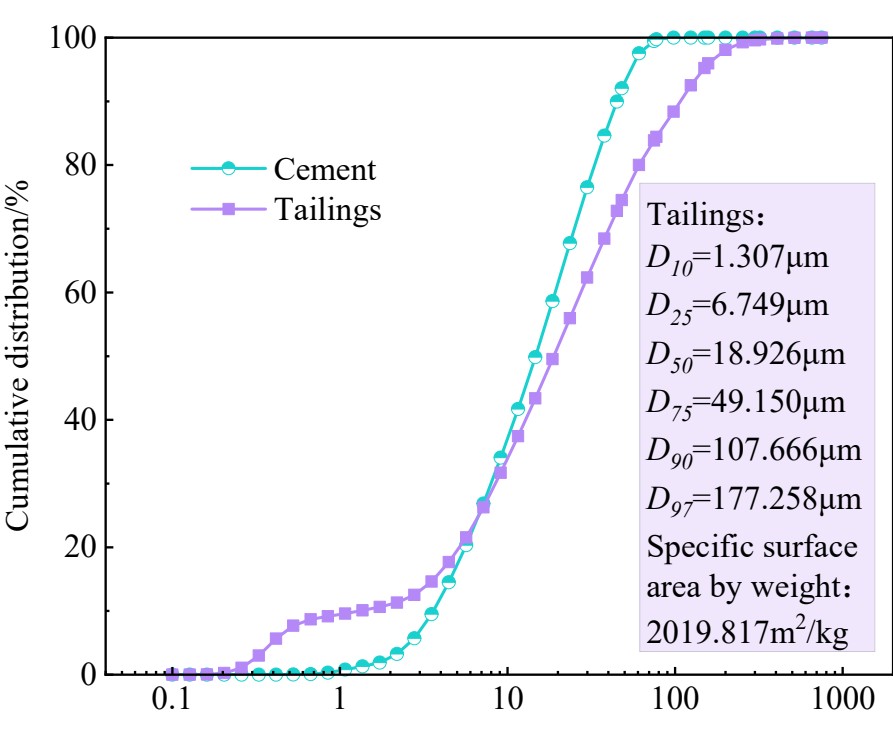

**Figure 1.** GSD curves for cement and gold tailings.

It has been presented that polypropylene fibers can effectively prevent the crack expansion of the cementitious tailings backfill material [56]. Thus, different lengths of polypropylene fibers were selected to prepare FRCTWR in this section. Table 1 presents polypropylene fiber's key properties.

**Table 1.** Key characteristics of polypropylene fiber employed in the backfill.

| Length (mm) | Density (g/cm³) | Tensile Strength (MPa) | Elongation Rate (%) | Elastic Modulus (GPa) |
|---|---|---|---|---|
| 6.0/12.0 | 5.9 | 386.0 | 3.8 | 27.2 |

### 2.2. Sample Preparation

To inspect the impact of longs/doses of PF fibers on CTWRF's strength evolution, specimens were designed with diverse fiber longs (6/12 mm) and doses (0-control, 0.3, 0.6, and 0.9 wt.%). The solid content of FRCTWR was 72 wt.%. The proportion of cement mass to total mass of tailings and waste rock was 1:8. Table 2 shows the ratio of each composition of specimens. Fiber content is the fraction of solid mass in CTWRF. PP6-0.3 indicates polypropylene fibers of 6 mm in length and 0.3 wt.% CTWRF. N-PP was non-reinforced CTWRF. Backfills were cast into a $50 \times 100$ mm² cylinder mold and then placed in a fixed temperature ($19 \pm 1$ °C) and humidity ($88 \pm 2\%$) conditioning chamber. The mold was removed after 48 h. Figure 2 shows the CTWRF's preparation processes.

**Table 2.** The proportions of each component of each specimen.

| Sample ID | Fiber Length (mm) | Fiber Content (wt.%) | Tailings Content (wt.%) | Waste Rock Content (wt.%) |
|---|---|---|---|---|
| PP6-0.3 | 6 | 0.3 | 60 | 40 |
| PP6-0.6 | 6 | 0.6 | 60 | 40 |
| PP6-0.9 | 6 | 0.9 | 60 | 40 |
| PP12-0.3 | 12 | 0.3 | 60 | 40 |
| PP12-0.6 | 12 | 0.6 | 60 | 40 |
| PP12-0.9 | 12 | 0.9 | 60 | 40 |
| N-PP | | | 60 | 40 |

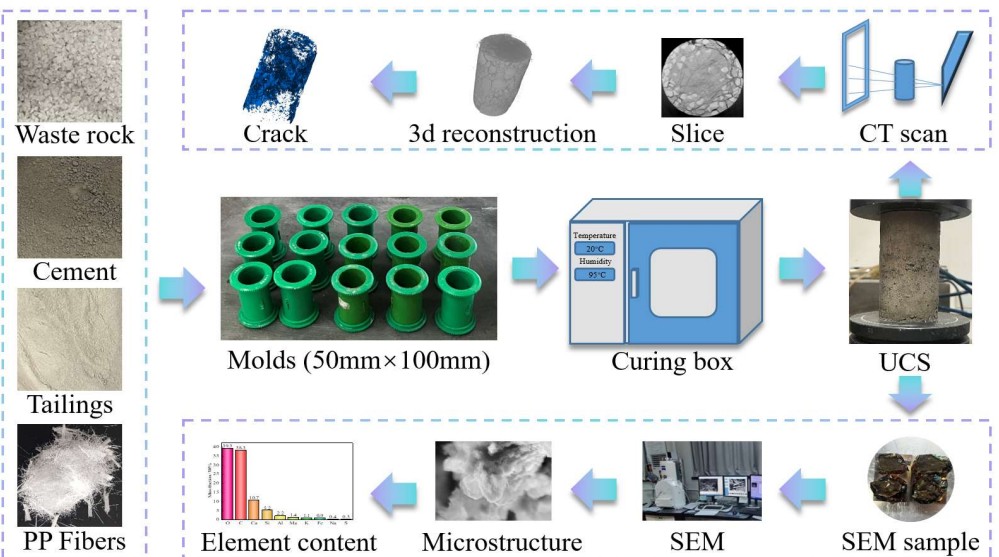

**Figure 2.** A flow chart of specimen preparation processes, including innovative materials/methods.

*2.3. Methods*

2.3.1. UCS Test

After CTWRF sample was cured for 7-day, the UCS experiment was conducted by a fully automatic mechanical press which entirely meets national standard GB/T16491-2008. The instrument model used for UCS testing was WDW-200D (POPWIL, Jinan, Shandong, China) and the loading range was 0–200 kN, the accuracy of mechanical tests was ±0.5%, the displacement measurement resolution was 0.001 mm, the displacement error is not more than 0.02 mm. A rate of 1 mm/min was also implemented during loading. Strength and stress–strain plots were automatically collected by PC. To find the average UCS value, a total of 3 experiments were made from each backfill mixture recipe.

2.3.2. X-ray CT Scanning Test

After completing the UCS experiments, the CTWRF specimens with the optimum fiber content in each fiber length were selected for industrial CT scanning tests, which were performed by high-resolution test equipment manufactured by YXLON, Hamburg, Germany. To lessen the interference of various factors during testing, two specimens were scanned simultaneously. The image size was 1024 × 1024. The pixel size was 80 μm. A total of 1097 slices were obtained for experiments.

2.3.3. SEM-EDS Measurement

Samples were prepared by selecting the fiber-containing areas in CTWRF matrices. Sample size was $1 \times 1 \times 1$ cm$^3$. To interrupt the hydration reaction of cemented materials, samples were kept in ethanol for 1-day. To enhance samples' electrical conductivity, samples dried for 12-h were sprayed with carbon twice by vacuum coating instrument. Specimens

were kept motionless using conductive tin foil tape. CTWRF's microstructure was observed by a Zeiss EVO 18 SEM (Zeiss, Oberkochen, Germany) which is used for imaging/analytical applications. The accelerating voltage was 20 kV. The hydration products within CTWRF specimens were also examined using an energy spectrometer.

## 3. Results and Discussion

### 3.1. Fiber Impact on FRCTWRF's Strength/Peak Strain Behavior

Figure 3a presents an evolution of UCS recorded for different CTWRF ratios. Note that the UCS value of N-PP was 0.89 MPa. However, the UCS value of CTWRF ranges from 0.67 to 0.93 MPa. The blending of fiber led to a small fluctuation in the strength of CTWRF, ranging from +4.49% and −24.72%. This is due to variations in fiber content and length. Overall, adding fibers gave rise to a decrease in CTWRF's strength. This is due to the higher aspect ratio and lower specific gravity of fiber, a property that contributes to the tendency of fibers to agglomerate and trapping of free water [55]. One can interpret that fibers can cause deterioration of pore structure in CTWRF, which was also a detrimental factor affecting the compressive strength of CTWRF [56]. UCS values of CTWRF were 0.77 to 0.93 MPa, and 0.83 to 0.67 MPa for a fiber long of 6 and 12 mm, respectively. CTWRF's strength was enhanced by 6 mm fiber long for the same fiber content. It is contrary to the results of fiber-doped tailings fill (raw materials are fiber, tailings, and cement) studied by Xue et al. [57]. This suggests that the mixing of waste rock affects the interaction mechanism among fiber, cement, and tailings. For CTWRF specimens having the same length of fibers, a rise in the content of fiber caused a growth and then a drop in UCS. The maximum strength of FRCTWRF was found for a fiber content of 0.6 wt.%.

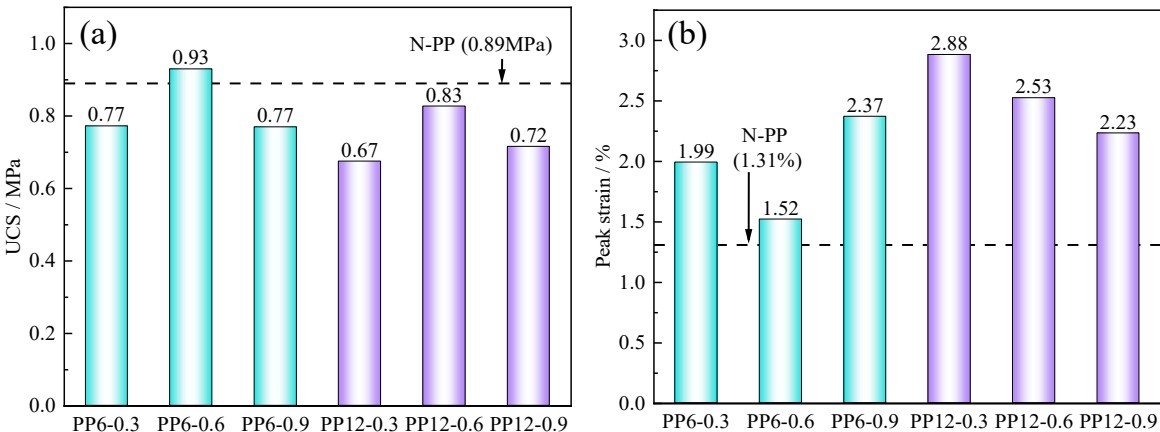

**Figure 3.** UCS and peak strain of FRCTWRF samples: (**a**) UCS; (**b**) peak strain. Cyan means PP6; purple means PP12.

Figure 3b shows the impacts of length and content of polypropylene fiber on CTWRF's peak strain. One can interpret that peak strain of N-PP is 1.31%. Adding fiber causes an increase in the peak strain of FRCTWRF by 16.03–119.85%, which is due to the good tensile properties of fibers. When exposed to external loading, fibers bonded between CTWRF matrix immersed more energy, which brings about a better deformability of CTWRF [58].

CTWRF's peak strains were 1.99% (0.3 wt.%), 1.52% (0.6 wt.%), and 2.37% (0.9 wt.%) for 6 mm fiber long, respectively. A rise in the content of fiber led to a decrease and then an increase in the peak strain of CTWRF. CTWRF's peak strains were 2.88% (0.3 wt.%), 2.53% (0.6 wt.%), and 2.23% (0.9 wt.%) for 12 mm fiber long. CTWRF's peak strain drops with growing fiber dosage. Overall, increasing the fiber length can lead to an increase in CTWRF's peak strain. Longer fibers have a larger contact area with CTWRF, resulting in a stronger bond. Longer fibers can boost the deformation capacity of CTWRF.

### 3.2. Fiber Effect on FRCTWRF's Stress–Strain/Toughness Behavior

Figure 4a shows the deformation or stress–strain plots of FRCTWRF and CTWRF. Note that RCTWRF's stress–strain plots are similar to those of CTWRF. The stress–strain curves are divided into four steps: (a) pore compacting step; (b) linear elastic step; (c) strain softening step; (d) crack extension step. It is easy to see that all curves have the same trend until the peak point. The influence of fibers on FRCTWRF's stress–strain curve exists with the post-peak stage. The load-bearing capacity of CTWRF decreases after getting peak strength. However, FRCTWRF's stress–strain plot declined in the level of post-peak, which indicated that fibers enlarged the post-peak load-carrying capacity and enhanced FRCTWRF's ductility. FRCTWRF curves were smoother for 12 mm fiber length than 6 mm fiber one, showing that longer fibers are more conducive to FRCTWRF ductility enhancement than shorter fibers.

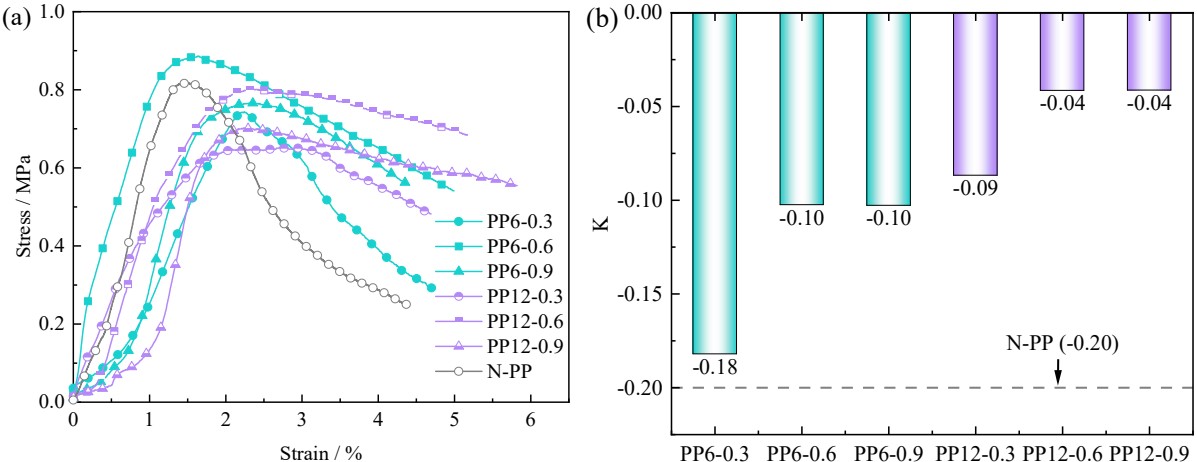

**Figure 4.** Stress–strain curves and slopes of post-peak curves for FRCTWRF specimens: (**a**) stress–strain plots; (**b**) slopes of post-peak curves.

Researchers were more concerned with the ductility enhancement of FRCTWRF by fibers than the strength change. To scrutinize the impact of fiber long/content on ductility of FRCTWRF, slope K of post-peak curve was used as a parameter for the strength evaluation of FRCTWRF. Figure 4b shows the impact of fiber long/dose on slope K. One can interpret that; the K value of N-PP is the smallest at −0.20. Increasing the fiber length can lead to enhanced post-peak ductility of FRCTWRF at a certain fiber content. Let us say that the K value for 6-0.3 is −0.18 and for 12-0.3 is −0.09, which is a 50% increase in K value. FRCTWRF with 0.3 wt.% fiber content has the minimum K value and the worst ductility for a confident fiber length. The K values increased by 44.4% (6 mm) and 55.6% (12 mm) for fiber content increasing from 0.3 wt.% to 0.6 wt.%, respectively. As the content of fiber continued to increase, the K value remained constant.

### 3.3. Microstructures Analysis Based on 3D Reconstruction

To probe the impact of fiber long variation on spatial distribution of fibers, waste rock, and fractures, FRCTWRF samples (PP 6-0.6 and PP 12-0.6) after UCS tests were tested by industrial CT scan technique. Figures 5 and 6 present the 2D slice images of 20 mm, 50 mm, and 80 mm from the bottom of the samples that were selected for comparative analysis. For visual observation, the 2D section images were pseudo-color enhanced using Image J software. Waste rock and fissures can be observed in 2D section images. Since the fiber gray values are similar to fissures, they can be distinguished by shape in the section images (filamentous structures are fibers in Figures 5 and 6).

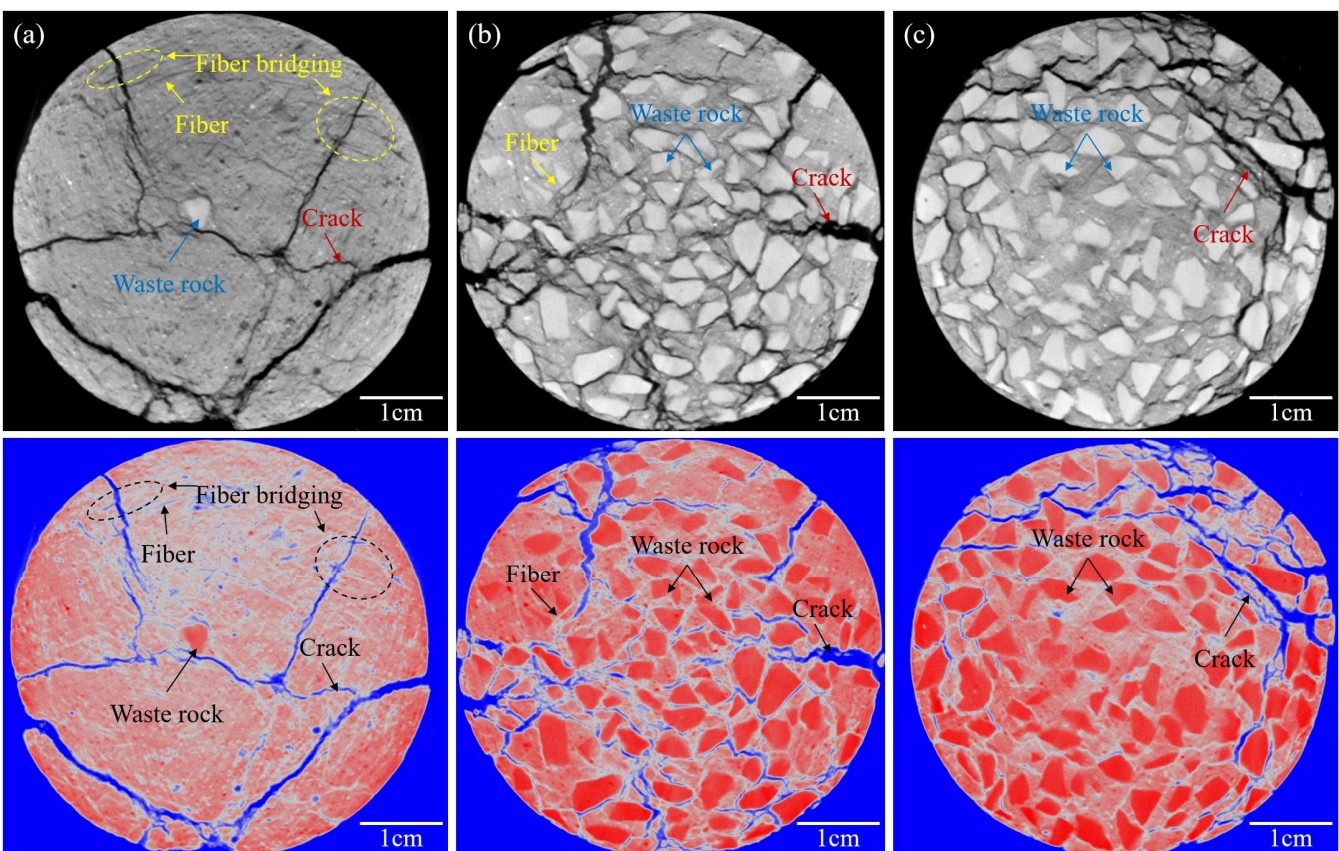

**Figure 5.** 2D CT images of PP6-0.6: (**a**) Z = 80 mm; (**b**) Z = 50 mm; (**c**) Z = 20 mm.

One can interpret from Figure 5 that 6 mm fiber lengths were distributed more uniformly in the FRCTWRF, and no fiber agglomeration was found. Some fibers penetrate both sides of the fissure and show an obvious bridging effect. The waste rock was not uniformly distributed in FRCTWRF, waste rock was less in the upper part of FRCTWRF (only one waste rock is found in Figure 5a), and waste rock was mostly focused on the middle/lower parts of FRCTWRF. Distribution of fissures and waste rocks was opposite, mainly distributed in the upper part of FRCTWRF. This may be related to the waste rock settlement [58].

One can interpret from Figure 6 that fibers of 12 mm in length are prone to agglomeration. Waste rocks are more abundant in Figure 6a,c and less abundant in Figure 6b. This shows that waste rocks are not uniformly distributed along the height. The section images also show a radially inhomogeneous distribution of waste rocks. This phenomenon is caused by the characteristics of waste rock itself. In Figure 6b, there is no waste rock distribution in the dense fiber region. This indicates that the inhomogeneous distribution of waste rocks is linked with fiber agglomeration. The fractures in PP12-0.6 specimen are only distributed at the edge of FRCTWRF composites due to the waste rock distribution. No cleavage similar to the one through the slice in Figure 5a was found.

To analyze the influence of fiber length variation on waste rock distribution, waste rock within FRCTWRF was extracted by the threshold segmentation. The percentage of waste rock area in the sliced images to the cross-sectional area of FRCTWRF was defined as waste rock percentage, and the waste rock percentage curve was used to quantitatively characterize the spatial distribution pattern of waste rock within FRCTWRF. Figures 7 and 8 showed the three-dimensional images of waste rock, waste rock percentage curves, and slice images for PP6-0.6 and PP12-0.6.

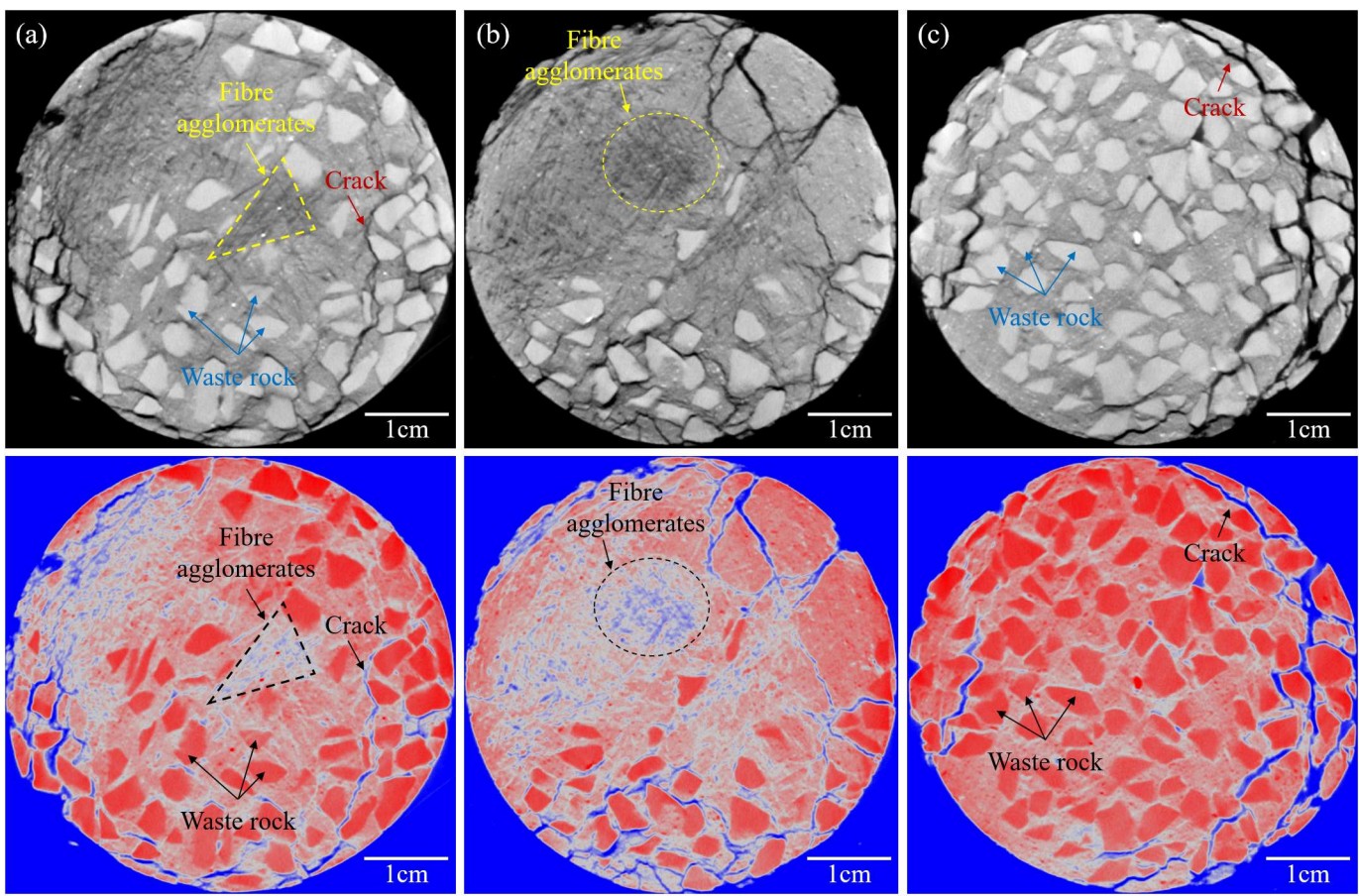

**Figure 6.** 2D CT images of PP12-0.6: (**a**) Z = 80 mm; (**b**) Z = 50 mm; (**c**) Z = 20 mm.

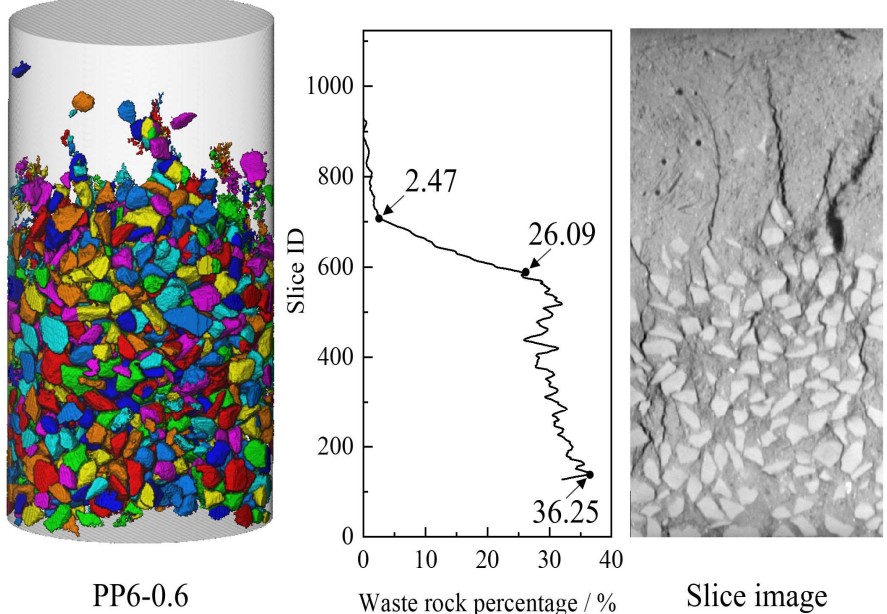

**Figure 7.** Distribution pattern of waste rock within PP6-0.6.

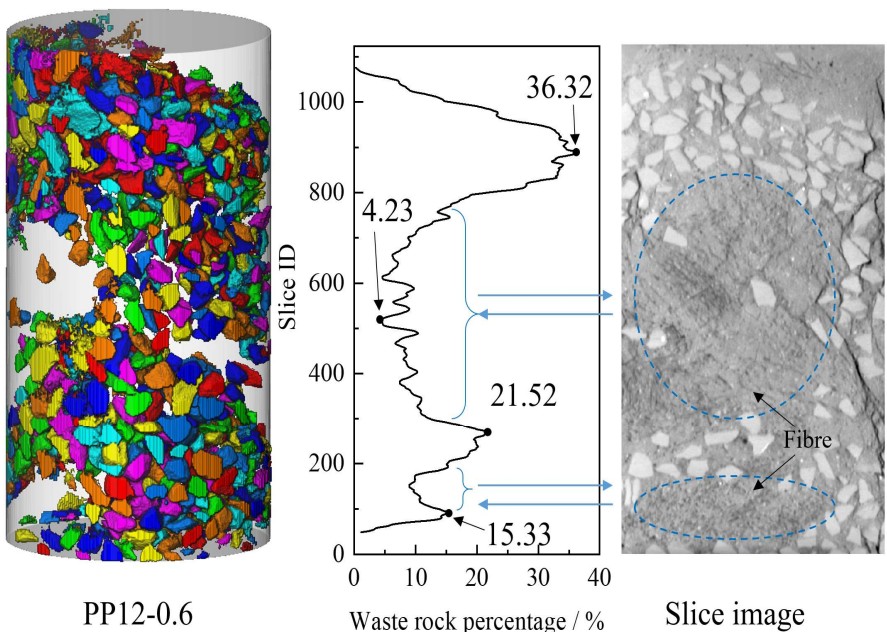

**Figure 8.** Distribution pattern of waste rock within PP12-0.6.

One can deduce from Figure 7 that waste rock percentage is less than 2.47 in the upper part of PP6-0.6. However, in the middle and lower part, the waste rock percentage was between 26.09 and 36.25. This indicated that waste rock is distributed in the middle and lower part of FRCTWRF, which was similar to the distribution pattern of coarse particles in none-fiber cementitious materials [59]. Thus, 6 mm fiber length has no effect on waste rock distribution. One can deduce from Figure 8 that there are three main peaks in waste rock distribution curve in PP12-0.6, which were 36.32, 21.52, and 15.33, respectively. Waste rocks were highly concentrated in the peak distribution. Combined with the section images, it could also be found that fibers are mainly distributed in two regions, which were between three peak points. This indicates that fiber agglomerates affect the waste rock distribution.

Figures 9 and 10 show 3D reconstructed images of cracks in PP6-0.6 and PP12-0.6, respectively. More tensile cracks are presented in PP6-0.6, which reveals tensile damage. The cracks are reduced in the bottom region, showing incomplete continuous sheet cracks. The crack structure is more complicated in PP12-0.6. The cracks showed discontinuous sheet shear cracks, and PP12-0.6 was shear damage. The crack volume was 4749.11 mm$^3$ in PP6-0.6 and 3764.69 mm$^3$ in PP12-0.6. One could say that increasing the fiber length resulted in a 20.73% drop in the crack volume. This shows that increasing fiber length can improve FRCTWRF's cracking resistance.

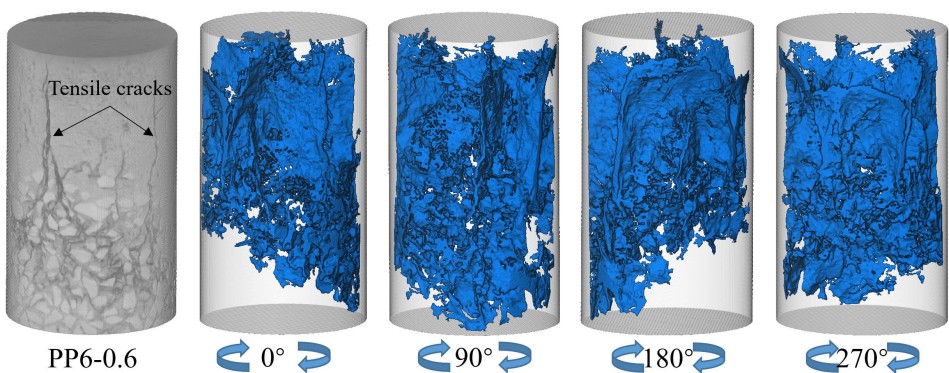

**Figure 9.** 3D reconstructed image of the cracks in PP6-0.6.

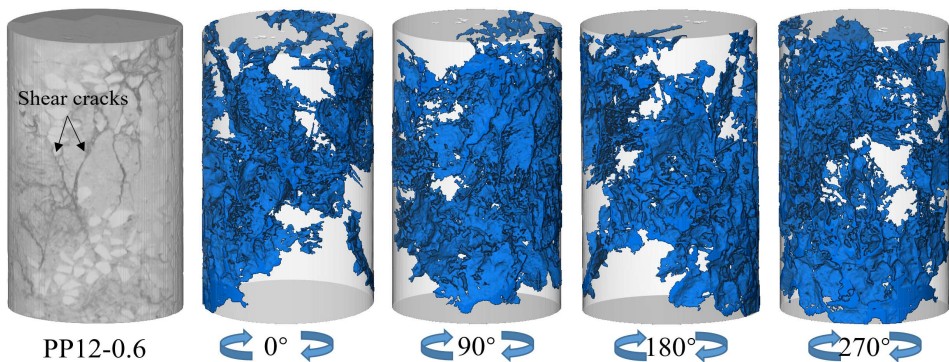

**Figure 10.** 3D reconstructed image of the cracks in PP12-0.6.

### 3.4. SEM Characteristics and Element Distribution

Figure 11 showed the microstructure images of PP6-0.6 and PP12-0.9. Three microstructures, namely FRCTWRF matrix, fibers, and pores, could be seen in SEM images. The FRCTWRF matrix was much denser. The pore diameters were 29.4 μm and 102.2 μm within PP6-0.6 and PP12-0.9, respectively. The fibers were connected to FRCTWRF by hydration products. A large quantity of hydration materials was involved in the external part of visible fibers. The main consequences displayed that some of hydration products were separated from the FRCTWRF matrix and attached to the surface of PF fibers after FRCTWRF cracking. This clearly indicates that the influence of bonding is well-observed between fibers and FRCTWRF matrix from SEM micrographs, and 12 mm fiber length was also deformed after being pulled out. In contrast, the 6-mm-long fibers were not significantly deformed.

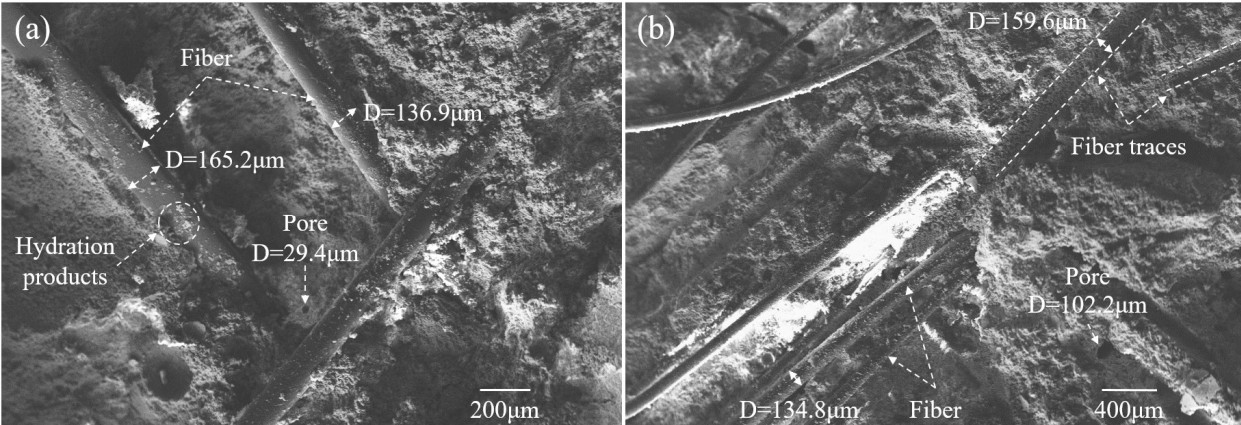

**Figure 11.** Microstructure image of FRCTWRF: (**a**) PP6-0.6; (**b**) PP12-0.9.

Figure 12 presents the morphological images of two extracted fibers in PP12-0.3. It could be observed that the pulled end of fiber was bent and deformed, while the other part was not deformed. A large number of neatly arranged scale-like FRCTWRF matrix were found on the surface of the extracted fibers. This was caused by the sliding of the fibers in the FRCTWRF matrix. This FRCTWRF matrix was not found on the surface of 6 mm long fibers. This may be due to the short sliding distance of 6 mm-long fibers in the FRCTWRF matrix.

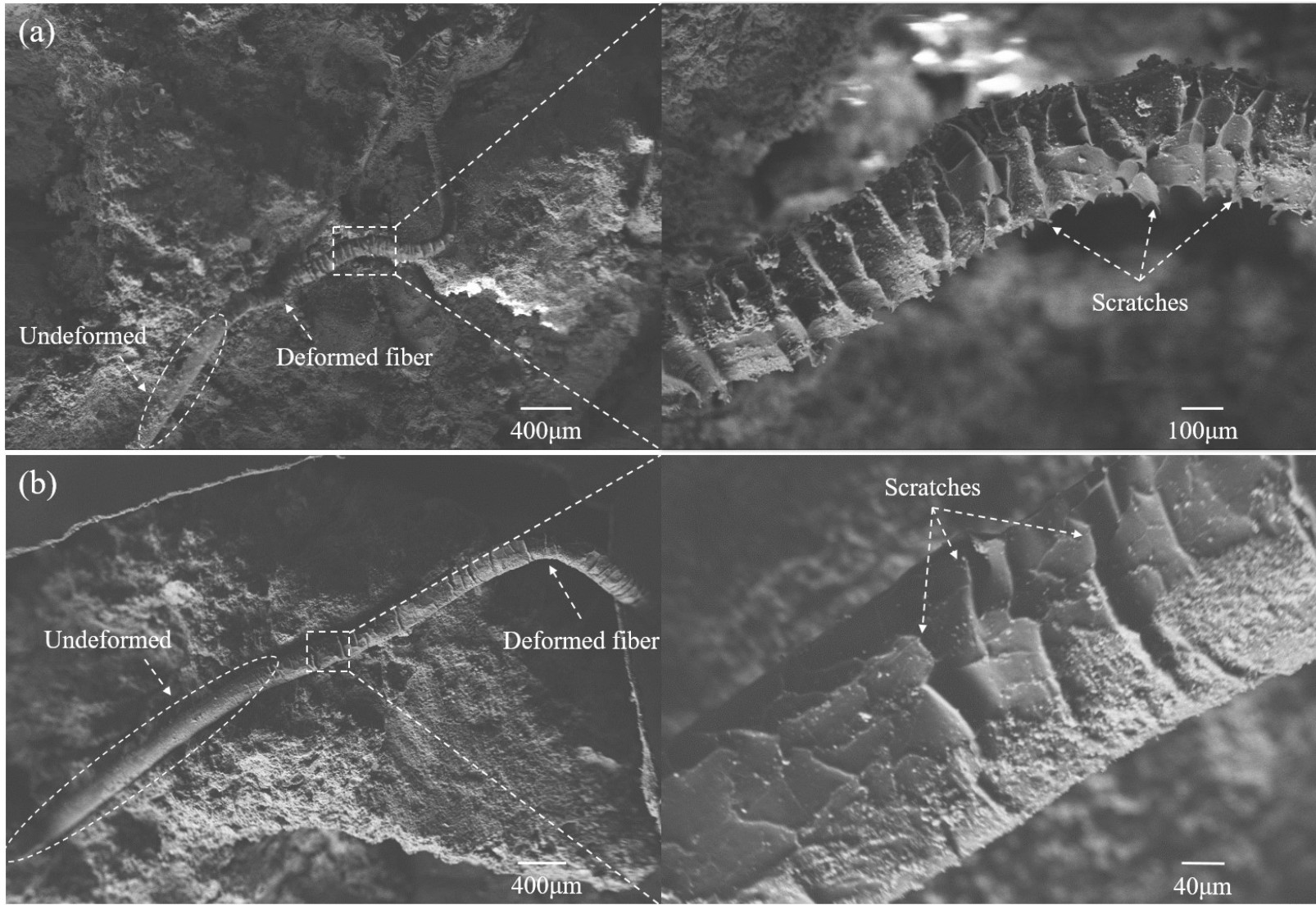

**Figure 12.** Microscopic image of bent and deformed fibers in PP12-0.3. (**a**) Area 1; (**b**) Area 2.

Figure 13 showed the microscopic morphology and elemental distribution of hydration materials within FRCTWRF. One could deduce that the main hydration materials were CSH gels, and a slight quantity of needle-like Ca alumina (AFt). From elemental distribution results, it can be seen that O, C, Ca, Si, and Al are the chief features of cement hydration products. Figure 14 showed the mass percentages of each element in hydration products. One can also say that O and C have the highest content with 39.3 wt.% and 38.3 wt.%, respectively.

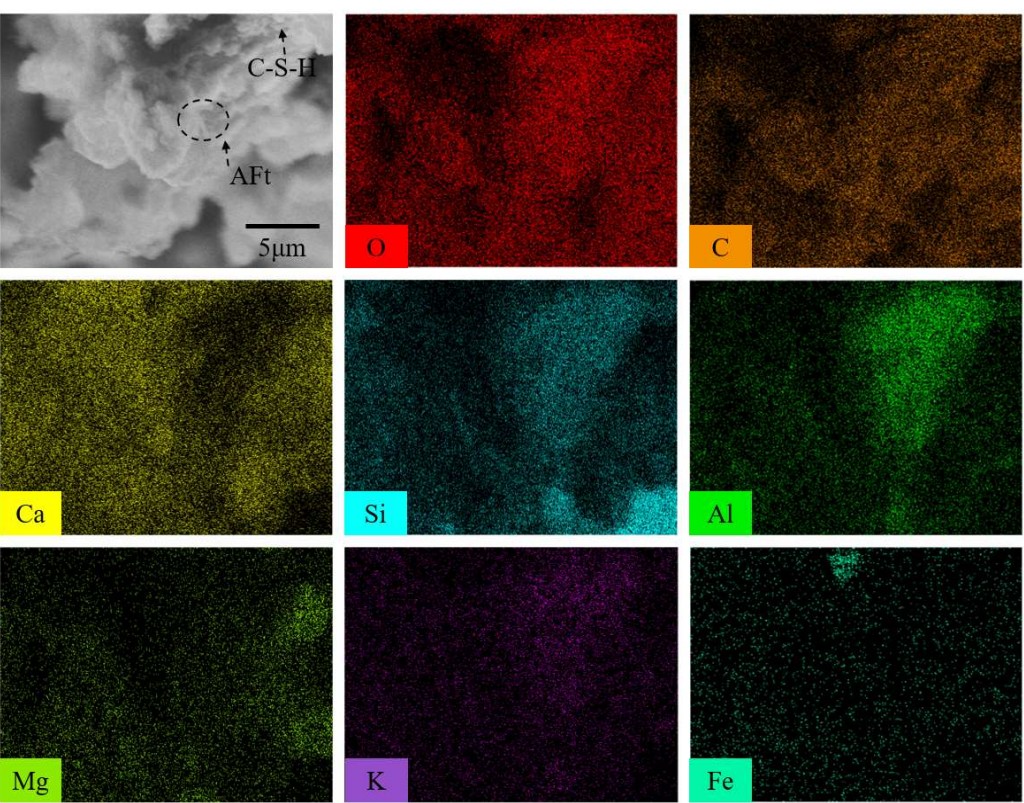

**Figure 13.** The main element mapping distributions of FRCTWRF sample.

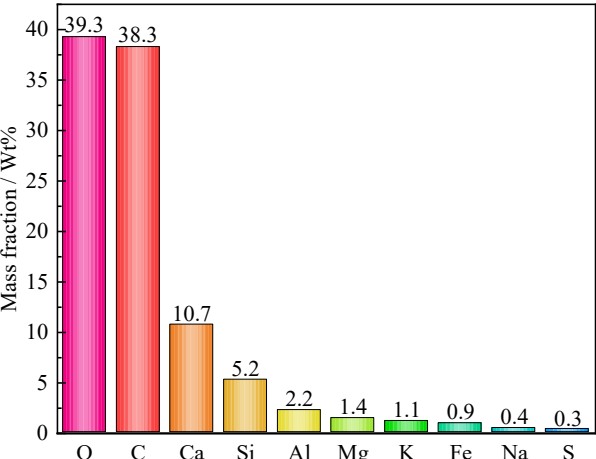

**Figure 14.** EDS values for hydrated products of FRCTWRF sample.

## 4. Conclusions

The impacts of fiber length/content on strength development of FRCTWRF were inspected by UCS tests. The internal fibers, waste rock, and fractures of FRCTWRF were visualized and analyzed by using a 3D reconstruction technique. The microstructure and

element distribution of FRCTWRF were investigated by SEM observations. From the above experiments, the following conclusions were drawn:

(1) Fiber incorporation can lead to changes in compressive strength of FRCTWRF. The changes in FRCTWRF's strength were +4.49% and −24.72%. A rise in the content of fiber led to an increase and then a drop in FRCTWRF's strength. The reinforcement effect of 6 mm long polypropylene fibers on FRCTWRF's strength was better than 12 mm long fibers. FRCTWRF's strength was maximum at 0.93 MPa for 6 mm long fiber and 0.6 wt.% fiber content.

(2) FRCTWRF's peak strains were all greater than those of CTWRF. The peak strains increased with rising fiber length. At 6 mm long fiber, a rise in the content of fiber led to a decrease and then an increase in peak strain. Peak strain decreases with growing fiber content for 6 mm long fiber. The maximum peak strain of FRCTWRF was 2.88% for 12 mm long fiber and 0.3 wt.% fiber content.

(3) Fibers governs the post-peak step of the FRCTWRF's stress–strain plot, and the post-peak ductility of FRCTWRF increases with increasing fiber length. 12 mm long fiber tends to agglomerate, and the fiber agglomeration affects the spatial distribution of waste stones. Waste rocks in PP6-0.6 are distributed in FRCTWRF's middle and lower parts. Waste rocks are uniformly distributed along the radial direction. Waste rocks in PP12-0.6 are not uniformly distributed along the axial and radial directions.

(4) Fibers have a bridging influence on cracks observed in FRCTWRF. The joint action of fibers and waste stones affects the damage mode of FRCTWRF, which is tensile damage in PP6-0.6 and shear damage in PP12-0.6. Increasing fiber long led to a drop of fracture body. FRCTWRF with 12 mm fibers is more resistant to cracking. Fibers and the FRCTWRF matrix were connected by hydration materials, which were typically CSH gel and needle-like Ca alumina. The key elements within the hydration products for filling specimens are O and C.

The present work offers the impact of fiber long/dose on strength and microstructure of cementitious backfills. It is thought that this new filling type, which is shaped by combining waste rock-fiber-tailings with different experimental methods, will be a trend research topic based on its cost-reducing and performance-enhancing properties. There is absolutely a need for more work in this area, but as such, the current work will contribute to other researchers by offering a diverse perspective and experimental methods. Some experiments are carried out by the same authors to examine the effect of some parameters on fresh/hardened performance of this fill type, and the results will be shared with the readers soon.

**Author Contributions:** Conceptualization, S.C.; methodology, B.G.; writing—original draft preparation, B.G.; writing—review and editing and supervision, E.Y. and S.C.; funding acquisition, S.C. All authors have read and agreed to the published version of the manuscript.

**Funding:** The writers frankly appreciate a generous fund granted from both the National Key Research and Development Program of China (No. 2022YFC2905004) and National Natural Science Foundation of China (No. 51804017).

**Data Availability Statement:** Not applicable.

**Conflicts of Interest:** The authors declare no conflict of interest.

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
