# Peer review of "Effect of Content and Length of Polypropylene Fibers on Strength and Microstructure of Cementitious Tailings-Waste Rock Fill"

_minerals, doi:10.3390/min13020142_

Round 1

Reviewer 1 Report

General Comment: 

In This paper titled "Influence of content and length of polypropylene fibers on  strength and microstructure evolution of cementitious tailings-waste rock fill " , The potential use of Polyprpylene fibers in mine backfilling was investigated. While the study looks interesting, the paper needs to be first improved before considering for publication. First of all, the manuscript requires proofreading to fix any grammatical issues and replace some of the words that were not appropriately used such as "pointedly", "solemn," and  "aptly". Below are some of the major comments on this manuscript:

Comments about the abstract:

The abstract is well written, but you need to start with a global statement about your research topic rather than speaking about your subject immediately. Please modify the abstract accordingly. 

Comments about section 1, "Introduction":

1.     Define ROM first in line 35. I assume it is Run of Mine Ore.

2.     Better to use "backfilled" rather than "re-filled" in line 39.

3.     lines 49 to 51. the phrase "could not only eliminate the safety hazards" and "green, low waste mining", How could you eliminate safety hazards by backfilling? a total elimination of safety hazards may be difficult to achieve in mining projects, but you can take some measures to control hazards. Please review the the Hierarchy of Controls to have a better understanding. There will be some hazards when backfilling is used too. Moreover, Backfilling may require the use of Cement and cement production is responsible for a huge amount of CO2 in the atmosphere. Therefore, the claim that backfill will promote "green mining" may not be appropriate. 

4. "dehydrated mine waste" in line 53, Please note that tailings do not necessarily need to be dried first to prepare backfill. They can be moist, and water is added based on moisture content. 

5.     line 55, "of CTB" only? What about "CPB"?

6.     line 57, replace the word "others" with some specific materials such as pozzolans, chemical or mineral additives, or supplementary cementitious materials.  

7.     Please review the statement from lines 60 to 62. How do fine-sized materials have higher porosity than coarse-sized materials? and the contradicting statement in line 76 indicates that the slighter the grain size, the higher the strength is.

8.     The word "scrutinize" in line 83 was not appropriately used here. I suggest replacing it with the word "investigate." 

9.     Lack of previous research reported on the subject; hence, the novelty needs to be clarified. 

Comments about section 2, "Materials and Methods”, 

1-    Define OPC first (line 92)

2-    Alternatively, you can say the GSD of gold tailings and cement are shown in figure 1 to avoid confusion. The current statement in line 95 may indicate they both have the same plot rather than they were both plotted on the same figure. They are not 100% identical.  

3-    Based on the parameters obtained from figure 1, indicates whether your tailings are poorly, well, or uniformly graded. 

4-    Cite the reference at the end of the first statement in line 101.

5-    line 113, do you mean 50X100 mm mold? it currently shows 50 100.

6-    how many samples were used for the UCS test for each mix? 

7-    No XRF or XRD was shown for tailings. Please add them to the paper. 

Comments about the results

Curing for 7 days may not be sufficient to judge the impact on UCS. Usually, we should see the effect after 28 days.

Final comments,

The study has some promising results, but what about the applicability of this technique, looking at it from different perspectives? For example, the availability of fibers, their costs, the challenge during mixing, etc.

Author Response

Reviewer #1:

Comments to the Author

General Comment:

In This paper titled "Influence of content and length of polypropylene fibers on strength and microstructure evolution of cementitious tailings-waste rock fill", The potential use of Polyprpylene fibers in mine backfilling was investigated. While the study looks interesting, the paper needs to be first improved before considering for publication. First of all, the manuscript requires proofreading to fix any grammatical issues and replace some of the words that were not appropriately used such as "pointedly", "solemn, "and "aptly". Below are some of the major comments on this manuscript:

Authors: We regret there were problems with the English. The manuscript has been carefully revised by a native English speaker to improve the grammar and readability.

Comments about the abstract:

The abstract is well written, but you need to start with a global statement about your research topic rather than speaking about your subject immediately. Please modify the abstract accordingly.

Authors: Thank you for your valuable suggestion. The authors have revised the abstract accordingly.

Comments about section 1, "Introduction":

  1. Define ROM first in line 35. I assume it is Run of Mine Ore.

Authors: Thank you for your valuable suggestion. The authors have defined it clearly in the revised manuscript.

  1. Better to use "backfilled" rather than "re-filled" in line 39.

Authors: Thank you for your valuable suggestion. The authors have use "backfilled" rather than "re-filled".

  1. lines 49 to 51. the phrase "could not only eliminate the safety hazards" and "green, low waste mining", How could you eliminate safety hazards by backfilling? a total elimination of safety hazards may be difficult to achieve in mining projects, but you can take some measures to control hazards. Please review the the Hierarchy of Controls to have a better understanding. There will be some hazards when backfilling is used too. Moreover, Backfilling may require the use of Cement and cement production is responsible for a huge amount of CO2 in the atmosphere. Therefore, the claim that backfill will promote "green mining" may not be appropriate.

Authors: Thank you for your valuable suggestion. The authors have corrected these inappropriate sentences.

  1. "dehydrated mine waste" in line 53, Please note that tailings do not necessarily need to be dried first to prepare backfill. They can be moist, and water is added based on moisture content.

Authors: Thank you for your valuable suggestion. The authors have revised it accordingly.

  1. line 55, "of CTB" only? What about "CPB"?

Authors: Thank you for your valuable suggestion. The authors have changed " of CTB " to " of CTB/CPB ".

  1. line 57, replace the word "others" with some specific materials such as pozzolans, chemical or mineral additives, or supplementary cementitious materials.

Authors: Thanks for your kind suggestion. The authors have changed " others " to " pozzolans ".

  1. Please review the statement from lines 60 to 62. How do fine-sized materials have higher porosity than coarse-sized materials? and the contradicting statement in line 76 indicates that the slighter the grain size, the higher the strength is.

Authors: Thanks for your kind suggestion. The authors have corrected this error in the revised manuscript.

  1. The word "scrutinize" in line 83 was not appropriately used here. I suggest replacing it with the word "investigate."

Authors: Thanks for your kind suggestion. The authors have changed " scrutinize " to " investigate ".

  1. Lack of previous research reported on the subject; hence, the novelty needs to be clarified.

Authors: Thanks for your kind suggestion. The authors have removed the word novelty.

Comments about section 2, "Materials and Methods”,

1- Define OPC first (line 92)

Authors: Thanks for your kind suggestion. The authors have defined the OPC.

2- Alternatively, you can say the GSD of gold tailings and cement are shown in figure 1 to avoid confusion. The current statement in line 95 may indicate they both have the same plot rather than they were both plotted on the same figure. They are not 100% identical.

Authors: Thanks for your kind suggestion. The authors have amended it to "the GSD of gold tailings and cement are shown in figure 1"

3- Based on the parameters obtained from figure 1, indicates whether your tailings are poorly, well, or uniformly graded.

Authors: Thanks for your kind suggestion. Based on the parameters obtained from figure 1, the tailings tested in this study are well graded.

4- Cite the reference at the end of the first statement in line 101.

Authors: Thanks for your kind suggestion. The authors have added a reference at the end of the first statement on line 101.

5- line 113, do you mean 50X100 mm mold? it currently shows 50 100.

Authors: Thanks for your kind suggestion. The authors have changed "50 100 " to " 50X100 ".

6- how many samples were used for the UCS test for each mix?

Authors: Thanks for your kind suggestion. Three samples were used for UCS tests for each mix.

7- No XRF or XRD was shown for tailings. Please add them to the paper.

Authors: Thanks for your kind suggestion. Please rest assured that we have not considered these analyses for the present manuscript, but will consider them in our future manuscripts incorporating mineralogical characterizations.

Comments about the results

Curing for 7 days may not be sufficient to judge the impact on UCS. Usually, we should see the effect after 28 days.

Authors: Thanks for your kind suggestion. The authors have made necessary explanations as follows:

An ongoing quality control/quality assurance (QC/QA) test program is crucial to ensure that the desired backfill strengths are achieved, at acceptable cement contents, without endangering the overall security of underground mining structures and operations. A curing time of 28 days, which allows the backfill matrix to sufficiently cure and reach a minimum compressive strength in order to ensure the safety of workers and the safe extraction of ores in the neighbor stopes of the backfilled area, are most often considered as part of a routine QC/QA test program in mines. However, the time is so critical in the mining industry, and as the mining cycle becomes shorter production increases significantly. Experiences show that, in the backfill mix made with fiber and OPC 42.5R cement, the cement hydration process starts abruptly, and strength gain begins immediately after final set. Accordingly, a 7-day curing time becomes sufficient for the backfill matrix which results in an equal or even more rapid gain in the strength

Final comments,

The study has some promising results, but what about the applicability of this technique, looking at it from different perspectives? For example, the availability of fibers, their costs, the challenge during mixing, etc.

Authors: We would like to thank you very much for giving us the opportunity to reply.

The applicability of this technique is possible in mining in terms of opportunities it offers despite some operational difficulties. Because fiber-reinforced backfills must be added at the end point, that is, at the point where the backfill is poured into the stope, not at the paste backfill plant above ground, otherwise it may cause blockages in the pipelines. Therefore, there is no possibility of mixing in the paste plant. Instead, the fibers are applied by adding them from a certain point on the backfilling area.

On the other hand, although the transportation of fibers from the surface to the underground and the increase in costs caused by it seems to be an operational and financial difficulty, its use will become quite widespread due to the significant opportunities it offers (high strength at lower cement amounts and providing more solid backfilling for both workers and underground structures). In fact, some mines in Europe are already successfully using this new backfill technique in their backfilling operations.

Reviewer 2 Report

The results obtained are valuable for the efficient use of FRCTWRF in mining practice. The figures and table are of good quality. The authors should be commended for this nice piece of work. The minor revisions below are suggested to improve the quality of the manuscript.

1. Table 1. All data is reserved to 1 decimal place.

2. Fig. 3. Sample ID have been described in Section 2.2, Ask the author to replace the abscissa with the sample ID

3. Fig. 4(b). Sample ID have been described in Section 2.2, Ask the author to replace the abscissa with the sample ID

4. L265. Please change “mm3” to “mm3”.

Author Response

Reviewer #2:

Comments to the Author

The results obtained are valuable for the efficient use of FRCTWRF in mining practice. The figures and table are of good quality. The authors should be commended for this nice piece of work. The minor revisions below are suggested to improve the quality of the manuscript.

Authors: We are very grateful to the reviewer for his/her careful and meticulous reading of our paper. The review is detailed and helpful to finalize our paper. We kindly acknowledge his/her constructive review.

  1. Table 1. All data is reserved to 1 decimal place.

Authors: Thanks for your kind suggestion. The authors have revised it accordingly.

  1. Fig. 3. Sample ID have been described in Section 2.2, Ask the author to replace the abscissa with the sample ID

Authors: Thanks for your kind suggestion. The author has changed the abscissa to sample ID.

  1. Fig. 4(b). Sample ID have been described in Section 2.2, Ask the author to replace the abscissa with the sample ID

Authors: Thanks for your kind suggestion. The author has changed the abscissa to the sample ID.

  1. L265. Please change “mm3” to “mm3”.

Authors: Thanks for your kind suggestion. The authors have changed " mm3" to " mm3" in the revised paper.

Reviewer 3 Report

The language of the text is very good and may need only minor revisions before publication. Similarly, the organization of the text and the sections are done in a well-structured manner. Some minor editorial comments are given for improvement:

- A more expanded introduction is required to mine backfill and the use of tailings and waste rock in it. This is a topic that has extensive publications going back 5-6 decades. In addition, the use of various cementing agents and their proportions should also be covered as an introduction to the use of fibers and their proportions so that a comparison can be made.

- The abstract is full of multiple acronyms, and looks very congested-confusing to the reader as there are multiple ones to remember. It is suggested that acronyms are used sparingly or the last two words are abbreviated only instead of having a 5- or 6-letter acronym (e.g., cemented TRF instead of CTWRF).

- Acronyms should be identified at their first mention and can only be used afterwards. For example, ROM on line 35 and OPC on line 92 are unknown words and have not been defined.

- Usage of "cylinder" is preferred to "tube-shaped" (line 113) as most backfill samples are prepared as cylinders. Therefore, this word is better known to the readers and is more common in the literature.

From a technical point of view, there are a few clarifications needed.

- Section 3.1 presents mixed results for the strength of the backfill. The variation is over 30% (from +5 to -25%) and line 155 indicates that the backfill has become weaker. Furthermore, this seems to be contradictory to the results obtained by Xue et al (line 159). More explanation is needed here as to what these results mean. Is it the effect of waste rock only or also of the fibers used in the backfill? What were the materials used by Xue et al?

- Section 3.3 discusses the segregation of waste rock in the backfill. This is a common occurrence in the use of cemented rock fill in underground mines as waste rock particles are too large to be homogenized with a mixer. It should be clarified if this segregation was only due to the waste rock presence or also due to fibers.

- From Section 3.2 onward, no mention of tailings is made in the results despite initially indicating that the backfill was made up of tailings and waste rock. Were the effects of tailings analyzed or was the focus only on waste rock? Also, there is no indication re the size distribution of the waste rock fragments, although it is mentioned they are between 5 and 7 mm (line 92). The distribution plays an important role especially when mixed with the fine tailings from a gold mine for which a distribution is given in Figure 1.

Author Response

Reviewer #3:

Comments to the Author

The language of the text is very good and may need only minor revisions before publication. Similarly, the organization of the text and the sections are done in a well-structured manner. Some minor editorial comments are given for improvement:

Authors: We would like to sincerely thank the reviewer for the positive feedback and for the thoughtful comments and constructive suggestions, which generously helped us to improve the quality of our manuscript.

- A more expanded introduction is required to mine backfill and the use of tailings and waste rock in it. This is a topic that has extensive publications going back 5-6 decades. In addition, the use of various cementing agents and their proportions should also be covered as an introduction to the use of fibers and their proportions so that a comparison can be made.

Authors: Thanks for your kind suggestion. The authors have extended the introduction section by taking into account all these aspects in the revised paper.

- The abstract is full of multiple acronyms, and looks very congested-confusing to the reader as there are multiple ones to remember. It is suggested that acronyms are used sparingly or the last two words are abbreviated only instead of having a 5- or 6-letter acronym (e.g., cemented TRF instead of CTWRF).

Authors: Thanks for your kind suggestion. In some backfill papers available in the literature, some abbreviations have already been created by many authors and are widely used. Of course, everyone acts differently at this point, as there is no common consensus on the backfill abbreviations. We hope that in a very short time there will be important developments in this field and paste backfill mechanics standards such as soil and rock mechanics (it differs from them) will emerge. In fact, we would like to state that there is such an attempt and that we have been working closely with this international team.

- Acronyms should be identified at their first mention and can only be used afterwards. For example, ROM on line 35 and OPC on line 92 are unknown words and have not been defined.

Authors: Thanks for your kind suggestion. The authors have explained " ROM " and " OPC " in the revised paper.

- Usage of "cylinder" is preferred to "tube-shaped" (line 113) as most backfill samples are prepared as cylinders. Therefore, this word is better known to the readers and is more common in the literature.

From a technical point of view, there are a few clarifications needed.

Authors: Thanks for your kind suggestion. The authors have changed " tube-shaped " to " cylinder " in the revised paper.

- Section 3.1 presents mixed results for the strength of the backfill. The variation is over 30% (from +5 to -25%) and line 155 indicates that the backfill has become weaker. Furthermore, this seems to be contradictory to the results obtained by Xue et al (line 159). More explanation is needed here as to what these results mean. Is it the effect of waste rock only or also of the fibers used in the backfill? What were the materials used by Xue et al?

Authors: Thanks for your kind suggestion. The authors have made necessary explanations as follows:

In this study, the waste rock content in fiber reinforced cementitious tailings-waste rock backfill is a constant value. These results suggest that the reduction in backfill strength is due to the fibers. Xue et al. prepared the backfill from polypropylene fiber and tailings and carried out an experimental study. The variation pattern between the strength of the backfill and the fiber content and length is inconsistent with the results obtained in this study. Therefore, the authors believe that the addition of waste rock to the fiber and tailings system resulted in the change of the fiber effect on the backfill.

- Section 3.3 discusses the segregation of waste rock in the backfill. This is a common occurrence in the use of cemented rock fill in underground mines as waste rock particles are too large to be homogenized with a mixer. It should be clarified if this segregation was only due to the waste rock presence or also due to fibers.

Authors: Thanks for your kind suggestion. The segregation of waste rock in backfill body is caused by waste rock.

- From Section 3.2 onward, no mention of tailings is made in the results despite initially indicating that the backfill was made up of tailings and waste rock. Were the effects of tailings analyzed or was the focus only on waste rock? Also, there is no indication re the size distribution of the waste rock fragments, although it is mentioned they are between 5 and 7 mm (line 92). The distribution plays an important role especially when mixed with the fine tailings from a gold mine for which a distribution is given in Figure 1.

Authors: Thanks for your kind suggestion. In this study, the authors have considered only the effect of content and length of fiber on the backfill performance and did not consider the effect of tailings.

Round 2

Reviewer 1 Report

The paper was improved and I recommend it for publication. Please make sure no errors appear in the references in the text. I can see some errors in different places in the manuscript.

Best wishes